# Regulating coordination number in atomically dispersed Pt species on defect-rich graphene for n-butane dehydrogenation reaction

Xiaowen Chen[1,2,9], Mi Peng[3,9], Xiangbin Cai[4,9], Yunlei Chen[5,6,9], Zhimin Jia[1,2], Yuchen Deng[3], Bingbao Mei[7], Zheng Jiang[7], Dequan Xiao[8], Xiaodong Wen[5,6], Ning Wang[4✉], Hongyang Liu[1,2✉] & Ding Ma[3✉]

Metal nanoparticle (NP), cluster and isolated metal atom (or single atom, SA) exhibit different catalytic performance in heterogeneous catalysis originating from their distinct nanostructures. To maximize atom efficiency and boost activity for catalysis, the construction of structure–performance relationship provides an effective way at the atomic level. Here, we successfully fabricate fully exposed $Pt_3$ clusters on the defective nanodiamond@graphene (ND@G) by the assistance of atomically dispersed Sn promoters, and correlated the n-butane direct dehydrogenation (DDH) activity with the average coordination number (CN) of Pt-Pt bond in Pt NP, $Pt_3$ cluster and Pt SA for fundamentally understanding structure (especially the sub-nano structure) effects on n-butane DDH reaction at the atomic level. The as-prepared fully exposed $Pt_3$ cluster catalyst shows higher conversion (35.4%) and remarkable alkene selectivity (99.0%) for n-butane direct DDH reaction at 450 °C, compared to typical Pt NP and Pt SA catalysts supported on ND@G. Density functional theory calculation (DFT) reveal that the fully exposed $Pt_3$ clusters possess favorable dehydrogenation activation barrier of n-butane and reasonable desorption barrier of butene in the DDH reaction.

[1] Shenyang National Laboratory for Materials Science, Institute of Metal Research, Chinese Academy of Sciences, Shenyang, P. R. China. [2] School of Materials Science and Engineering, University of Science and Technology of China, Shenyang, P. R. China. [3] Beijing National Laboratory for Molecular Sciences, College of Chemistry and Molecular Engineering and College of Engineering, and BIC-ESAT, Peking University, Beijing, P. R. China. [4] Department of Physics and Center for Quantum Materials, Hong Kong University of Science and Technology, Clear Water Bay, Kowloon, Hong Kong SAR, P. R. China. [5] State Key Laboratory of Coal Conversion, Institute Coal Chemistry, Chinese Academy of Sciences, Taiyuan, P. R. China. [6] University of Chinese Academy of Science, Beijing, P. R. China. [7] Shanghai Institute of Applied Physics, Chinese Academy of Sciences, Shanghai, P. R. China. [8] Center for Integrative Materials Discovery, Department of Chemistry and Chemical Engineering, University of New Haven, West Haven, CT, USA. [9]These authors contributed equally: Xiaowen Chen, Mi Peng, Xiangbin Cai, Yunlei Chen. ✉email: phwang@ust.hk; liuhy@imr.ac.cn; dma@pku.edu.cn

Heterogeneous catalysis plays an indispensable role in chemical production[1]. For a typical supported metal catalyst, many factors including the crystallographic surface, chemical composition, particle size, and metal–support interaction can affect catalytic performances[2]. Recently, it has been found that well-dispersed nanoparticles (NPs), clusters, and isolated metal atoms (or single atoms, SAs) exhibited surprisingly different catalytic performances from bulk materials with the development of the advanced characterization tools and well-controlled synthesis technology, leading to the establishment of the structure–performance relationship at the atomic level[3–11].

In several previous works, SAs catalysts exhibit advantages such as unique reaction pathway[12], low adsorption energy of reactants/intermediate[13] and maximal atom utilization[4,14,15]. Nevertheless, it is worth noting that clusters, though with inactive bulk components, were more active than SAs in some catalytic reactions. Anderson et al.[16] deposited size-selected $Au_n^+$ ($n = 1$, 2, 3, 4, 7) on $TiO_2$ support to study the relationship between Au size and CO oxidation activity. Owing to adverse CO-Au binding, Au or $Au_2$ was inactive for CO oxidation. The activity was found in the order of $Au_7 > Au_3 > Au_4 > Au_2 \approx Au_1$. Corma et al.[17,18] studied the size effect of different types of Pt species (single atoms, clusters, and nanoparticles) supported on various oxides. In the low-temperature NO reduction reaction with CO, the surface of Pt clusters favored NO dissociation and CO oxidation. The moderate adsorption of CO on Pt clusters could suppress catalyst poisoning, leading to higher activity than that of Pt SAs. Szanyi et al.[19] found that Ru clusters favored $CH_4$ formation in contrast to Ru single atoms. Due to the limited ability to activate hydrogen, single Ru atoms can only allow CO formation but cannot further hydrogenate it to $CH_4$. However, we find later that the modulation of the chemical state of metal species by strong metal–support interaction is more important for observed selectivity regulation (metallic Ir particles for $CH_4$ while partially oxidized Ir species for CO production) in $CO_2$ hydrogenation reaction over $Ir/CeO_2$ catalyst with different size of Ir species[20]. Nevertheless, the highly dispersed subnanometer-sized metal clusters (<1 nm) achieve maximum atomic exposure and utilization and possess various chemical coordination environments and chemical states that eventually affect the activity, selectivity, and stability of catalysts. It is the cluster with certain size, instead of single atoms, that was linked to high stability, selectivity, and activity in the heterogeneous catalysis, where the catalysts showed remarkable performance on structure-sensitive catalytic reactions[21,22]. But the small clusters always suffered from aggregating into metal NPs at high temperature, limiting their applications into high-temperature reactions[23–26]. Therefore, considering industrial application, designing thermodynamically stable metal cluster with low metal loading (especially noble metal) still remains highly desired.

Direct dehydrogenation (DDH) for light olefins production has been a thematic research area resulting from the energy shortage and petroleum gas upgrading scenario[27,28]. The primary catalyst designated for DDH is a Pt-based catalyst[29,30]. The side reactions of DDH including coke deposition, hydrogenolysis, and cracking are usually structure-sensitive, which prefer to occur on Pt NPs with large average Pt–Pt coordination numbers (CN)[31]. An ideal solution to suppress side reactions and promote catalytic performance is to downsize Pt NPs and regulate CN of Pt to a moderate level. In general, the addition of promoter metals to Pt afforded bimetallic and alloying systems, which can improve both the dispersion and stability for Pt species. For instance, as to n-butane dehydrogenation, Pt particle sizes decreased with increasing Sn loading in the $PtSn/\theta$-$Al_2O_3$ catalyst. Higher atomic efficiency and the formation of PtSn alloying phases were correlated with higher activity and n-$C_4^{2-}$ selectivity. But the conversion rate remained low at each active site in such a large size PtSn alloying system[32]. Considering maximize atomic utilization, Pt/Cu single atom alloy catalyst is proposed in propane dehydrogenation. Compared with Pt NPs, isolated Pt atoms dispersed on copper nanoparticles dramatically enhance the propylene selectivity and stability[33]. Several bimetallic cluster catalysts have also been reported, including small raft-like PtSn clusters[34], PtSn/ND@G[35], Pt/Sn-Beta catalysts[36], PtSn cluster[37,38], and PtZn clusters defined in zeolites[39,40], which displayed high activity for DDH reaction with excellent durability. But it also remains major challenges to synthesize clusters with precisely controlled metal-metal coordination numbers, and fundamentally understand the structure effects (SA, cluster and nanoparticle) on dehydrogenation performance at atomic level.

In this paper, we fabricated fully exposed $Pt_3$ clusters stabilized on the defective graphene through Pt–C bond, with the geometric partitioning by the atomically dispersed Sn promoter, which can precisely tune the CN of supported Pt clusters based our previous reported methods[35]. The obtained $Pt_3$ clusters with 0.5 wt% Pt loading showed higher conversion for n-butane DDH than Pt NPs and Pt SAs supported on ND@G. At a relatively low temperature (450 °C), n-butane rate of $Pt_3$ clusters has achieved 1.138 mol·$g_{Pt}^{-1}$·$h^{-1}$. Moreover, we systemically established the structure–performance relationship by correlating the DDH activity with the average CN of Pt–Pt bond on ND@G supported Pt NP, Pt cluster, and Pt SA catalysts. By density functional theory (DFT) calculations, we found that the $Pt_3$ cluster catalyst's unique structure facilitates the activation of C–H bond and the desorption of butene. Such a structure–performance relationship may provide an insight for rationally designing highly active heterogeneous DDH catalysts in atomic scale.

## Results

**Construction of $Pt_3$ clusters by the assistance of Sn**. In order to fabricate ND@G supported $Pt_3$ cluster catalysts, a serial of PtSn/ND@G catalysts were prepared by the co-impregnation method, denoted as $Pt_{0.8Sn}$/ND@G (Sn/Pt atomic ratio = 0.85), $Pt_{1.7Sn}$/ND@G (Sn/Pt atomic ratio = 1.7), $Pt_{3.4Sn}$/ND@G (Sn/Pt atomic ratio = 3.4), and $Pt_{6.8Sn}$/ND@G (Sn/Pt atomic ratio = 6.8). The Pt and Sn precursors anchored on the ND@G support composed of a diamond core and a defect-rich graphene shell (see the TEM images in Supplementary Fig. 1). The detailed structure and morphology of ND@G has been described in our previous reports[12,35,41]. The reference Pt/ND@G catalyst without the addition of any Sn was prepared by the same procedure. To elucidate the detailed structure of as-prepared PtSn catalysts, the study by aberration-corrected high-angle annular dark-field scanning transmission electron microscopy (HAADF-STEM) was carried out. Due to the difference in Z-contrast, the supported Pt species can be easily distinguished from Sn atoms, as shown in Fig. 1. As a control, the obtained Pt NPs of Pt/ND@G (nanoparticle diameter, d = $2.81 \pm 0.95$ nm) were located on the ND@G surface (Supplementary Fig. 2a-f), and the lattice fringes of the Pt NPs were apparent, implying the good crystallinity of the as-prepared Pt NPs on ND@G. The lattice spacing of Pt NPs is 0.23 nm, corresponding to the (111) facet of typical Pt NPs. Notably, with the addition of 0.25 wt% Sn, the well-dispersed Pt species, as nanoparticles with d = $1.43 \pm 0.33$ nm, were observed in $Pt_{0.8Sn}$/ND@G (Supplementary Fig. 2g-i). The atomically dispersed Sn highlighted by green circles were located around the small Pt NPs (Supplementary Fig. 2i), suggesting that the presence of Sn can dramatically promote the dispersion of Pt NPs. For $Pt_{1.7Sn}$/ND@G, the uniformly dispersed Pt clusters highlighted by red circles were found as closely connected "islands" on the ND@G support, surrounded by atomically dispersed Sn

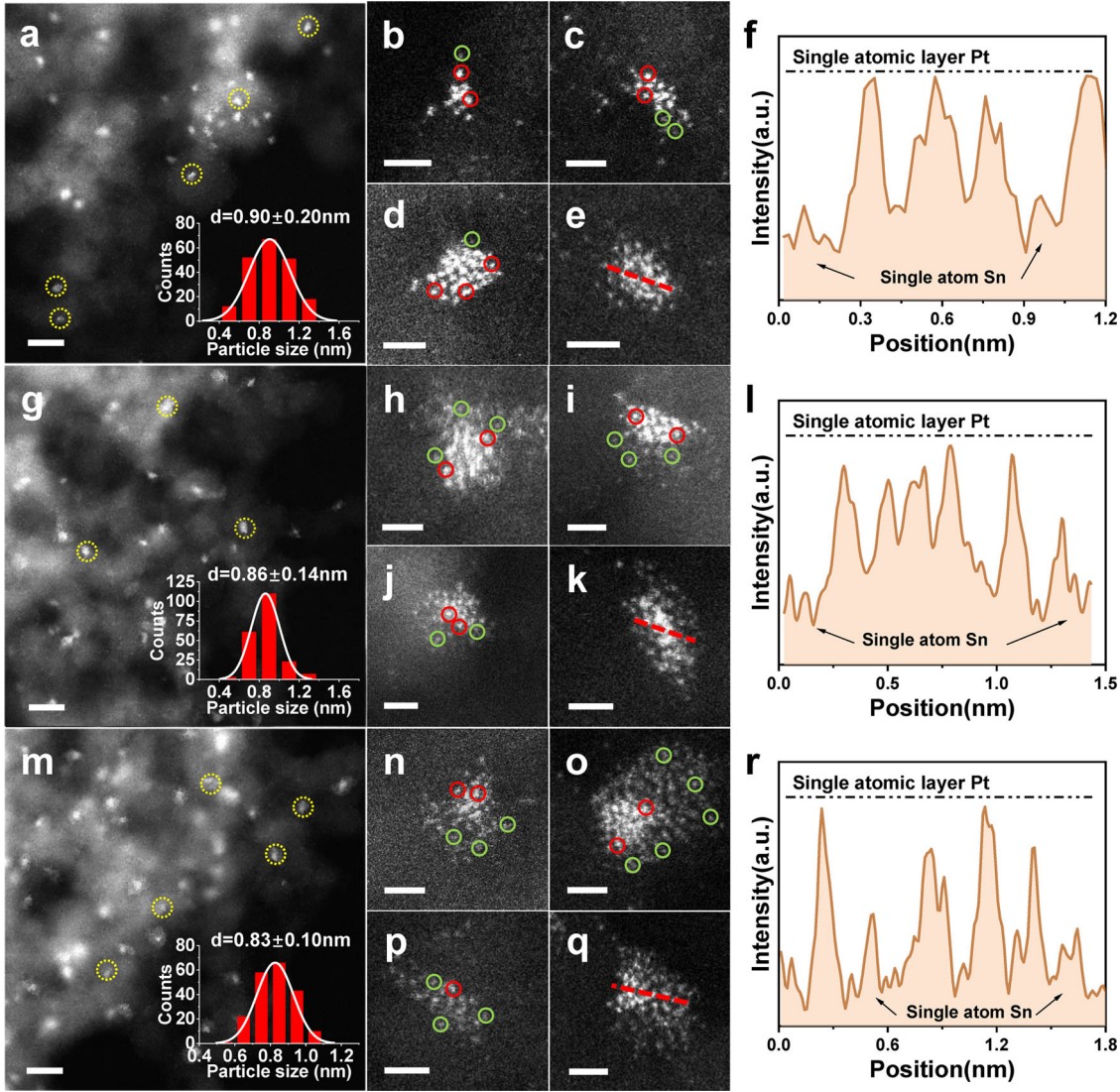

**Fig. 1 Microscopic characterizations of Pt$_{1.7Sn}$/ND@G and Pt$_{3.4Sn}$/ND@G and Pt$_{6.8Sn}$/ND@G. a–e** HAADF-STEM images of Pt$_{1.7Sn}$/ND@G. **g–k** HAADF-STEM images of Pt$_{3.4Sn}$/ND@G. **m–q** HAADF-STEM images of Pt$_{6.8Sn}$/ND@G. In the images, Pt clusters are highlighted by the red circles, and atomically dispersed Sn atoms are highlighted by the green circles. **f, l, r** are the extracted line profiles along red directions in **e**, **k**, **q**, demonstrating the pronounced intensity difference between Pt and Sn, consistent well with their distinct atomic numbers (Z), together with the single-atomic-layer thickness of a typical Pt cluster. Scale bars: **a**, **g**, **m**, 5 nm; **b–e**, **h–k**, and **n–q**, 1 nm.

species in Fig. 1a-e. The extracted line profile shows that the Pt clusters were monolayered (Fig. 1f). It suggested that no Pt atoms covering each other and all Pt atoms were fully exposed[35]. Interestingly, as to the Pt$_{3.4Sn}$/ND@G and Pt$_{6.8Sn}$/ND@G catalysts, with the further addition of Sn, Pt atoms remained highly dispersed as irregular-shape tiny Pt clusters. Multiple adjacent Pt clusters aggregated into "islands", which could be clearly resolved on the ND@G support (Fig. 1g–k and m–q). The extracted line profile showed that Pt clusters remained one-atomic-layer thick (Fig. 1l and r). However, high-density atomically dispersed Sn species were observed around the Pt clusters, resulting from larger loading amounts of Sn. These results indicate that the size and structure of Pt species are related to the geometric partitioning effect of Sn atoms, indicating that mono-dispersed Sn species facilitate the formation of atomically dispersed Pt clusters. When the atomic ratio Sn/Pt was greater than 1.7, almost all Pt atoms existed in the form of atomically dispersed Pt clusters.

Besides, from the XRD profiles (Supplementary Fig. 3), the diffraction peak at 39.7° corresponding to the (111) plane of Pt

crystal was found in Pt/ND@G, which is in good agreement with the STEM observation. There was no prominent diffraction peak observed for Pt$_{0.8Sn}$/ND@G, which confirms the formation of well-dispersed tiny Pt NPs in Pt$_{0.8Sn}$/ND@G. In terms of Pt$_{1.7Sn}$/ND@G, Pt$_{3.4Sn}$/ND@G, and Pt$_{6.8Sn}$/ND@G, no diffraction peak of Pt NPs was detected, indicating that Pt clusters were atomically dispersed on the surface of ND@G, consistent well with HAADF-STEM results. To further reveal the unique structure of the catalysts, the Pt dispersion state was determined by $H_2/O_2$ titration measurements. For the Pt$_{1.7Sn}$/ND@G catalyst, the dispersion of Pt was as high as 99.1%, indicating that almost all the Pt atoms were fully exposed on Pt$_{1.7Sn}$/ND@G. It means that all the Pt atoms on Pt$_{1.7Sn}$/ND@G are available for adsorptions under the reaction conditions. Meanwhile, excess Sn species partly covered the Pt atoms on Pt$_{3.4Sn}$/ND@G (the dispersion of Pt was 81.9%) and Pt$_{6.8Sn}$/ND@G (the dispersion of Pt was 5.6%). In other words, the excess Sn species did not further promote the Pt dispersion but covered up the Pt clusters, preventing Pt atoms from being exposed to adsorbed reactant molecules.

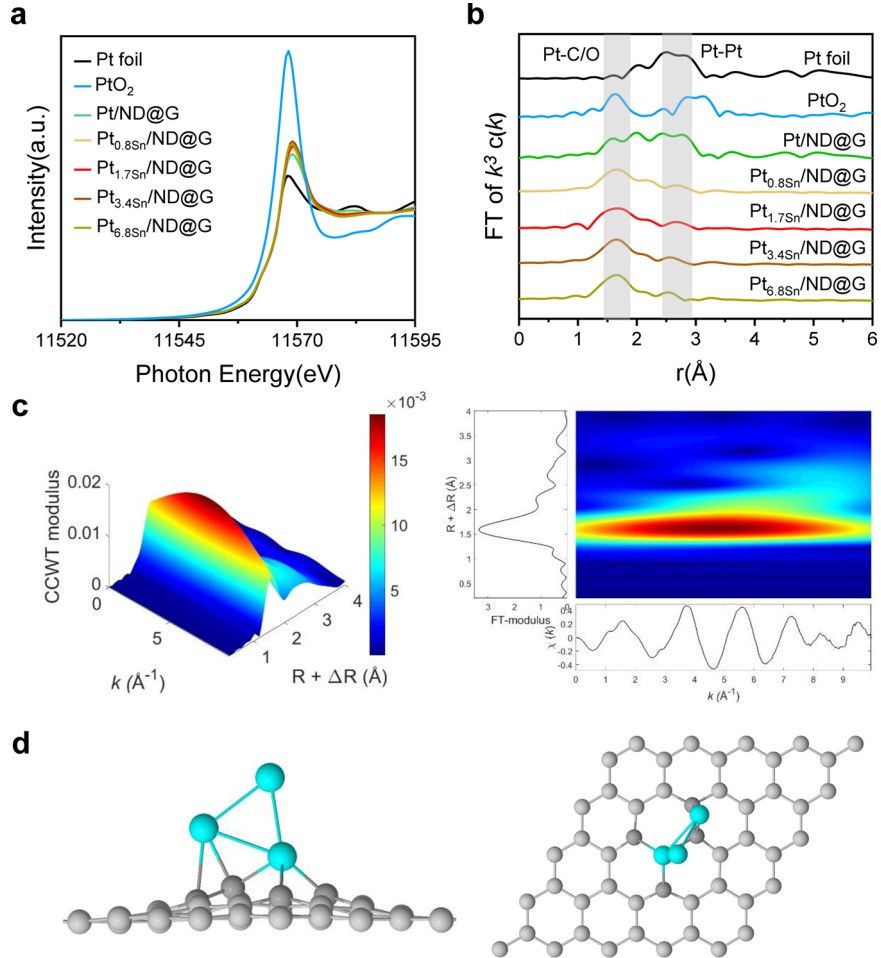

**Fig. 2 Synchrotron XAFS measurements of catalysts. a** Pt L$_3$ XANES spectra of above as-prepared catalysts, Pt foil and PtO$_2$. **b** $k^3$-weighted EXAFS spectra of above as-prepared catalysts, Pt foil and PtO$_2$. **c** WT analysis of Pt$_{1.7Sn}$/ND@G. **d** Optimized structure of Pt$_3$ cluster embedded into graphene (Pt$_3$-Gr, through Pt−C bond) from top and side views.

**Table 1 Pt L$_3$-edge EXAFS fitting results for as-prepared catalysts.**

| Sample | Type | CN | R(Å) | σ²(×10⁻³Å²) | ΔE (eV) | R-factor |
|---|---|---|---|---|---|---|
| Pt foil | Pt−Pt | 12 | 2.77 | | | |
| Pt/ND@G | Pt−C/O | 3.7 | 2.07 | 9.0 | 7.0 | 1.43% |
| | Pt−Pt | 6.0 | 2.73 | 6.0 | 2.3 | |
| Pt$_{0.8Sn}$/ND@G | Pt−C/O | 3.1 | 2.05 | 4.7 | 3.1 | 2.00% |
| | Pt−Pt | 3.2 | 2.65 | 9.0 | −4.1 | |
| Pt$_{1.7Sn}$/ND@G | Pt−C/O | 4.0 | 2.07 | 7.0 | 5.5 | 1.50% |
| | Pt−Pt | 2.3 | 2.68 | 6.1 | 0.3 | |
| Pt$_{3.4Sn}$/ND@G | Pt−C/O | 4.3 | 2.07 | 6.8 | 4.6 | 1.52% |
| | Pt−Pt | 2.0 | 2.62 | 6.7 | −3.4 | |
| Pt$_{6.8Sn}$/ND@G | Pt−C/O | 4.0 | 2.07 | 5.6 | 4.6 | 0.91% |
| | Pt−Pt | 2.1 | 2.65 | 8.0 | −1.5 | |

The X-ray adsorption fine structure (XAFS) measurement was employed to provide detailed information of the structure and the local environment of Pt and Sn species. From the X-ray absorption near edge structure (XANES) spectroscopy, the intensity of the white line for as-prepared catalysts situated above that of Pt foil, indicating the existence of slightly positively charged Pt$^{\delta+}$ species formed by stabilized on the defect-rich ND@G support (Fig. 2a). The EXAFS spectra of the as-prepared samples (Pt/ND@G, Pt$_{0.8Sn}$/ND@G, Pt$_{1.7Sn}$/ND@G, Pt$_{3.4Sn}$/ND@G, and Pt$_{6.8Sn}$/ND@G), Pt foil, and PtO$_2$ are shown in Fig. 2b. The detailed EXAFS fitting parameters for these catalysts

are shown in Table 1. For the Pt$_{0.8Sn}$/ND@G catalyst, it exhibited a distinct peak at 1.7 Å and a weak peak at 2.6 Å, which matched up with the first coordination shell of Pt−C/O and Pt−Pt, respectively, indicating that Pt NPs were located on ND@G support through Pt−C bonds. The average CN of Pt−C/O was 3.1, and the average CN of Pt−Pt was 3.2. Notably, for the Pt$_{1.7Sn}$/ND@G, Pt$_{3.4Sn}$/ND@G and Pt$_{6.8Sn}$/ND@G samples, they all showed a strong signal of Pt−C/O and a relatively weak signal of Pt−Pt. For the Pt$_{1.7Sn}$/ND@G catalyst, the average CN of Pt−Pt is ~2 (2.3), verifying the presence of Pt$_3$ clusters. Due to the limitation of the characterization methods (especially X-ray absorption spectroscopy (XAS)), the coordination property of Pt atoms is statistically averaged. The realistic Pt clusters on current catalyst could have distribution in both atomicity and configuration[42], but the majority of Pt clusters compose of around three Pt atoms. Moreover, all the Pt atoms in the Pt$_3$ clusters (dispersion of Pt was 99.1%) are fully exposed. Significantly, both the features of high dispersion and fully exposure allow all the Pt atoms accessible for adsorbing reactant molecules. For the Pt$_{3.4Sn}$/ND@G and Pt$_{6.8Sn}$/ND@G catalyst, the CN of Pt−Pt is 2.0 and 2.1, respectively, similar to that of Pt$_{1.7Sn}$/ND@G, suggesting that Pt atoms in Pt$_{3.4Sn}$/ND@G and Pt$_{6.8Sn}$/ND@G mostly existed in the form of Pt$_3$ clusters. In contrast, for the Pt/ND@G catalyst, the average CN of Pt−Pt bond was 6, indicating the formation of large Pt NPs, agreeing well with the STEM results. The wavelet transformation (WT) of Pt L$_3$-edge

EXAFS oscillations visually displayed the structure of Pt species in both the k and R spaces (Fig. 2c and Supplementary Fig. 5). Figure 2c is the WT contour plot of $Pt_{1.7Sn}$/ND@G, showing a Pt–C/O back-scattering contribution near 1.6 Å. Moreover, another minor peak at 2.6 Å in $Pt_{1.7Sn}$/ND@G can be attributed to the Pt–Pt scattering, further verifying the presence of Pt clusters with low CN. The Fourier transform of $k^3$-weighted EXAFS at Sn K-edge was performed to examine the coordination environment of Sn atoms. As shown in Supplementary Fig. 6, all the samples show an apparent peak at 1.5 Å that corresponds to the first coordination shell of Sn–C or Sn–O, and no Sn–O–Sn or Sn–Sn scattering was observed, indicating the formation of atomically dispersed Sn species. No Pt–Sn bonding was observed, suggesting that Pt and Sn are not in the form of Pt–Sn alloy. The WT of Sn K-edge EXAFS oscillations displayed only a strong signal at 1.5 Å that corresponds to the Sn–C/O coordination (Supplementary Fig. 9), verifying that Sn was atomically dispersed in all the samples. To further confirm the local coordination environment of Pt clusters, the optimized structure of $Pt_3$ cluster by density functional theory calculations is shown in Fig. 2d. A triangular $Pt_3$ cluster is anchored on the ND@G via Pt–C bonds. The mean bond length of Pt–C and Pt–Pt bonds is 2.05 Å and 2.56 Å, respectively, which is in good agreement with the EXAFS measurements in experiment. As shown in Supplementary Fig. 10, the simulated STEM image also shows the monolayered $Pt_3$ clusters on defective graphene surface, in agreement well with the experimental STEM observations. Supplementary Fig. 11 illustrates the relationship between Pt–Pt CN (or Pt dispersion) and Sn/Pt atomic ratio. Notably, with the increase of Sn/Pt atomic ratio from 0 to 1.7, the Pt–Pt CN decreases (and the Pt dispersion increases) in general, suggesting that mono-dispersed Sn species play a vital role in promoting the dispersion of Pt species on ND@G support. However, when the Sn/Pt atomic ratio was beyond 1.7, the Pt–Pt CN kept almost unchanged with the increasing atomic ratio of Sn/Pt. The Pt dispersion decreased to 5.6% at Sn/Pt = 6.8, indicating that the excess amount of Sn did not reduce the size of Pt clusters, but surrounded or covered up the Pt clusters, as shown by the STEM results.

The n-butane DDH reaction was evaluated over Pt/ND@G, $Pt_{0.8Sn}$/ND@G, $Pt_{1.7Sn}$/ND@G, $Pt_{3.4Sn}$/ND@G, and $Pt_{6.8Sn}$/ND@G to understand the role of Sn species on catalytic performance under the atmospheric condition and at 450 °C. The conversion of n-butane and the selectivity to $C_4$ olefin over those catalysts are shown in Fig. 3a and Supplementary Table 2. As shown in Fig. 3a, Pt/ND@G was initially active, and then a fast deactivation of selectivity from 11.6 to 8.2% in 10 h was observed. The n-butane conversion rate over the Pt/ND@G catalyst was only 0.373 mol·$g_{Pt}^{-1}$·$h^{-1}$. The value of $K_d$ (deactivation rate constant, see its expression in the "reaction analysis" section) and the initial selectivity at 10 h were 0.0421 $h^{-1}$ and 96.0%, respectively. In general, the deactivation of DDH reaction was attributed to the sintering of Pt NPs with large Pt–Pt CN, resulting in structure-sensitive side reactions and coke formation to block Pt active sites on the catalyst surface[43]. Notably, benefited from the mono-dispersed Sn, the catalytic performance of $Pt_{0.8Sn}$/ND@G was dramatically enhanced. The butane conversion reached 25.9% and then dropped to 20.8% in 10 h test. The selectivity towards $C_4$ olefin reached 98.9% at the initial stage. The value of $K_d$ was 0.0313 $h^{-1}$, showing that the lifetime was extended with decreasing CN of Pt–Pt. Significantly, the $Pt_{1.7Sn}$/ND@G catalyst possessed excellent catalytic activity and remarkably high selectivity for n-butane DDH. Figure 3a shows that the n-butane conversion reached up to 35.4% and still exhibited a high activity level of 30.9% after 10 h reaction. The selectivity towards $C_4$ olefin was as high as 99.0% at the initial stage, and the value for $K_d$ was only 0.0223 $h^{-1}$. When further adding Sn species, the

conversion and n-butane rate dramatically decreased over the $Pt_{3.4Sn}$/ND@G and $Pt_{6.8Sn}$/ND@G catalysts as shown in Fig. 3a, b. However, the $C_4$ olefin selectivity over $Pt_{3.4Sn}$/ND@G and $Pt_{6.8Sn}$/ND@G were both close to that of $Pt_{1.7Sn}$/ND@G. For $Pt_{3.4Sn}$/ND@G and $Pt_{6.8Sn}$/ND@G, the structure of atomically dispersed $Pt_3$ was almost intact, but the excess Sn species covered up the $Pt_3$ clusters, causing Pt dispersion decrease from 99.1% to 5.6%, as shown in Fig. 3a and Supplementary Fig. 11. Only a fraction of Pt atoms was available for adsorption during the reaction. Therefore, we conclude that the excess amount of Sn promoter for $Pt_3$ clusters can significantly reduce the conversion and n-butane rate. As shown in Fig. 3b, with the decreasing CN of Pt–Pt, the n-butane conversion rate increased firstly for $Pt_{0.8Sn}$/ND@G and $Pt_{1.7Sn}$/ND@G. When Sn/Pt = 1.7, the atomically dispersed $Pt_3$ cluster was thought to be the optimum catalyst, as all the Pt atoms are full exposed for n-butane DDH. Moreover, with the increase of reaction time, the butane conversion and the selectivity for $Pt_{1.7}$Sn/ND@G catalyst remained constant at 24% even after 50-h of reaction (see Fig. 3c). Additionally, HAADF-STEM images of $Pt_{1.7Sn}$/ND@G after 10-h in n-butane DDH showed that the fully exposed Pt clusters remained atomically dispersed on ND@G, implying their good stability during the DDH reaction (Supplementary Fig. 12).

**Size effect of Pt NP and cluster and SA in butane DDH.** To gain insights into the catalyst structure dependence for n-butane DDH reaction, the structure–performance relationship in catalysis was further established by comparing three representative catalysts: Pt/ND@G (Pt NP), $Pt_{1.7Sn}$/ND@G ($Pt_3$ cluster), and $Pt_1$/ND@G (Pt SA). Firstly, aberration-corrected HAADF-STEM was employed to investigate the morphology of $Pt_1$/ND@G. As shown in Supplementary Fig. 13a and b, individual Pt atoms without any visible clusters or NPs are exhibited as bright dots in high contrast to the ND@G support. The XRD patterns of $Pt_1$/ND@G showed that no Pt crystal or Pt-oxide phase was observed (Supplementary Fig. 13c). In the Pt $L_3$-edge EXAFS spectra (Supplementary Fig. 13d), the sample showed a dominant peak at 1.7–1.8 Å, which can be ascribed to the coordination of Pt atom to light elements such as C or O. Clearly, no Pt–Pt bond was observed, indicating that all the Pt atoms were atomically dispersed on ND@G support, consistent with the HAADF-STEM observation. WT of Pt $L_3$-edge EXAFS (Supplementary Fig. 13e) further revealed the atomic dispersion nature of $Pt_1$/ND@G₀

Figure 4 and Supplementary Table 3 summarize the reaction data of Pt/ND@G (Pt NP), $Pt_{1.7Sn}$/ND@G ($Pt_3$ Cluster), and $Pt_1$/ND@G (Pt SA), respectively. The fully exposed $Pt_3$ clusters showed the best catalytic performance for n-butane DDH reaction than Pt/ND@G and $Pt_1$/ND@G. Clearly, the n-butane DDH rate of $Pt_{1.7Sn}$/ND@G (1.138 mol·$g_{Pt}^{-1}$·$h^{-1}$) was higher than those of Pt/ND@G (0.373 mol·$g_{Pt}^{-1}$·$h^{-1}$) and $Pt_1$/ND@G (0.193 mol·$g_{Pt}^{-1}$·$h^{-1}$). Consistently, the initial selectivity towards butene on $Pt_1$/ND@G was only 97.1%, followed by $Pt_{1.7Sn}$/ND@G (96.6%) and Pt/ND@G (93.1%). Figure 4b illustrated the correlation between catalytic activity and coordination number. Generally, dehydrogenation catalysts with small CN have shown high efficiency resulting from their high utilization of metal atoms[31]. But interestingly, the n-butane conversion rate did not increase proportionally with a decrease of the Pt–Pt CN in Fig. 4b. A maximum butane conversion rate occurred for $Pt_{1.7Sn}$/ND@G, but $Pt_1$/ND@G.

To better understand the relationship between metal size and its catalytic performance, DFT calculations were conducted to unveil the difference in dehydrogenation among Pt/ND@G, $Pt_{1.7Sn}$/ND@G, and $Pt_1$/ND@G. In the DFT calculations, Pt (111) surface, Pt SA on single-lay graphene ($Pt_1$-Gr), and a triangle $Pt_3$

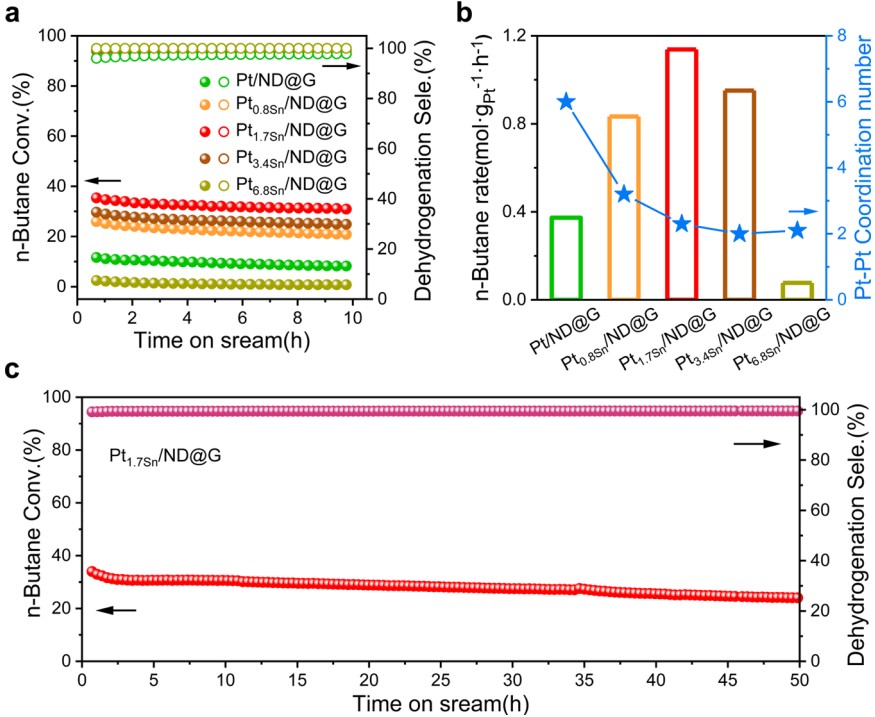

**Fig. 3 Catalytic performance for n-butane DDH. a** n-butane conversion and C$_4$ olefin selectivity of Pt/ND@G, Pt$_{0.8Sn}$/ND@G, Pt$_{1.7Sn}$/ND@G, Pt$_{3.4Sn}$/ND@G and Pt$_{6.8Sn}$/ND@G. **b** the conversion rate of n-butane and Pt–Pt CN. **c** stability test over Pt$_{1.7Sn}$/ND@G at 450 °C.

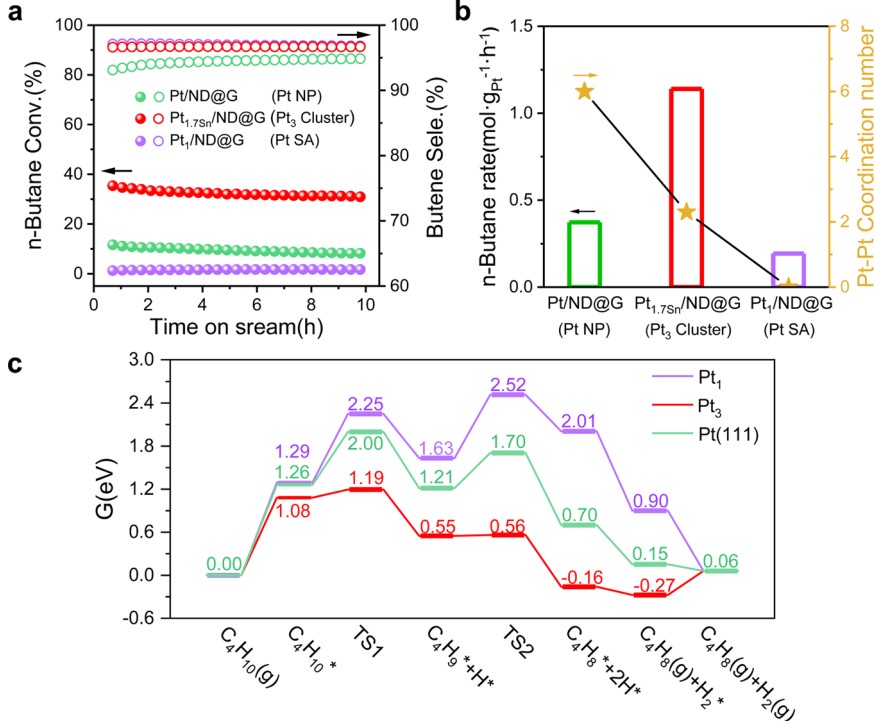

**Fig. 4 Catalytic performance for n-butane DDH. a** n-butane conversion and butene selectivity by time-on-stream during n-butane DDH at 450 °C, GHSV = 18000 mL · g$_{cat}$$^{-1}$ · h$^{-1}$, n-C$_4$H$_{10}$:H$_2$ = 1:1 with He balance. **b** the conversion rate of n-butane in the n-butane DDH and Pt–Pt CN. **c** energy profile of butane dehydrogenation to 2-butene on the Pt$_1$-Gr, Pt$_3$-Gr, and Pt (111). (purple: Pt$_1$-Gr, red: Pt$_3$-Gr, green: Pt (111)).

cluster doped into single-vacancy graphene (Pt$_3$-Gr) were used to model Pt/ND@G, Pt$_1$/ND@G, and Pt$_{1.7Sn}$/ND@G, respectively. The dehydrogenation process follows four steps: (i) the adsorption of n-butane; (ii) the dehydrogenation of n-butane to surface adsorbed 2-C$_4$H$_9$ species (2-C$_4$H$_9$*); (iii) the surface adsorbed 2-

C$_4$H$_8$ (2-C$_4$H$_8$*) is generated via the further dehydrogenation of 2-C$_4$H$_9$*; (iv) the desorption of 2-C$_4$H$_8$* to form the product 2-butene gas. Supplementary Tables 4–6 summarize the reaction energy and barriers of butane dehydrogenation for Pt$_1$-Gr, Pt$_3$-Gr, and Pt (111). The Gibbs free energy profile of butane

dehydrogenation to 2-butene on these three models is shown in Fig. 4c. One can see that the overall barrier of butane dehydrogenation are 2.52 eV (Pt$_1$-Gr), 1.19 eV (Pt$_3$-Gr), and 2.00 eV (Pt (111)), respectively, suggesting that the Pt$_3$-Gr is the most active for butane dehydrogenation. These data provide a rational interpretation for the high catalytic activity of Pt$_3$-Gr and low activity of Pt$_1$-Gr in our experimental observations. Deep hydrogenation (e.g., further dehydrogenation of 2-butene here) is known to be the origin of coke and hydrogenolysis[44]. The difference between the energy barriers of deep dehydrogenation ($E_{DH}$) and the desorption ($E_{DP}$) of 2-butene ($\Delta E_S = E_{DH} - E_{DP}$) can be used to evaluate the selectivity of dehydrogenation from alkanes to alkenes[44]. The more positive of the value $\Delta E_S$ indicates the better selectivity of the catalyst. Supplementary Fig. 20 provides the $\Delta E_S$ for Pt$_1$-Gr (0.45 eV), Pt$_3$-Gr (0.11 eV), Pt (111) (−0.04 eV), respectively. Due to insufficient metal active sites to the further dehydrogenation and weak adsorption of 2-butene, Pt$_1$-Gr exhibits the best selectivity compared to Pt$_3$-Gr and Pt (111). Based on the calculated results, Pt$_3$-Gr is predicted to have high activity and selectivity for the butane dehydrogenation to butene, while Pt$_1$-Gr is expected to have high selectivity and relatively low activity, which agrees well with the aforementioned experimental observations.

## Discussion

By optimizing the loading amount of Sn promoter, we fabricated fully exposed Pt$_3$ clusters where the atomically dispersed Sn atoms play the role of geometric partitioning. We constructed the structure–performance relationship between Pt NP, fully exposed Pt$_3$ cluster, and Pt SA for n-butane DDH reaction. The fully exposed Pt$_3$ clusters showed the highest n-butane conversion and remarkable alkene selectivity, compared to Pt NPs and Pt SAs, resulting from the facilitated activation of C–H bond and the desorption of butene. Such relationship between Pt CN and n-butane DDH activity provides a valuable insight in the structure effect on catalytic performance and thus a new avenue to design DDH catalysts with high activity, selectivity, and stability.

## Methods

**Materials**. Nanodiamond (ND) powders with the average diameter of 30 nm were purchased from Beijing Grish Hitech Co., China. Analytical grade chloroplatinic acid (H$_2$PtCl$_6$·6H$_2$O) and Tin (II) chloride dehydrate (SnCl$_2$·2H$_2$O) as metal precursors were purchased from Sinopharm Co. Ltd.

**Catalyst preparation**. The nanodiamond@graphene (ND@G) hybrid carbon support was prepared by annealing fresh nanodiamond powders at 1100 °C (heating rate 5 °C·min$^{-1}$) for 4 h under flowing Ar gas (80 mL·min$^{-1}$). When it finished and cooled to room temperature, the final powder, ND@G, was collected for further use. A series of PtSn/ND@G catalysts were prepared by co-impregnation method using a fixed weight loading of Pt (0.5%) with varying weight loading of Sn: Pt$_{0.8Sn}$/ND@G (Sn/Pt atomic ratio = 0.85), Pt$_{1.7Sn}$/ND@G (Sn/Pt atomic ratio = 1.7), Pt$_{3.4Sn}$/ND@G (Sn/Pt atomic ratio = 3.4), Pt$_{6.8Sn}$/ND@G (Sn/Pt atomic ratio = 6.8). First, certain amount of H$_2$PtCl$_6$ (20 g/L) and SnCl$_2$·2H$_2$O (6.0 g/L) were dissolved in 1 ml ethanol and generated a clear solution. Then, 100 mg ND@G powder was added and impregnated in the liquid solution. After that, the samples were dried in air at 80 °C for another 6 h. Finally, the solid sample were calcined in Ar (80 mL · min$^{-1}$) at 500 °C for 4 h and subsequently reduced in H$_2$ gas (80 mL · min$^{-1}$) at 500 °C for 1 h. For reference, the Pt/ND@G (0.5 wt% Pt) was prepared with the same process without the addition of Sn. Pt$_1$/ND@G (0.1 wt % Pt) was also prepared with the similar process without the addition of Sn and reduced by thermal treatment in H$_2$ gas at 200 °C for 1 h.

**Characterizations**. The series of samples were characterized by X-ray diffraction (XRD) on a D/MAX-2500 PC X-ray diffractometer with monochromated Cu K radiation ($\lambda = 1.54$ Å). HAADF-STEM measurements were conducted with a JEOL JEM ARM 200CF aberration-corrected scanning transmission electron microscope at 200 kV accelerating voltage. XAFS measurements of the samples were carried out in Shanghai Synchroton Radiation Facility (SSRF). The H$_2$-O$_2$ titration measurements were performed on a Micromeritics AutoChem II 2920 equipped with a thermal conductive detector.

**Reaction analysis**. The catalytic test for the DDH reaction of n-butane was tested using a fixed-bed stainless-steel micro-reactor with a quartz lining under atmosphere pressure at 450 °C, equipped with an online gas chromatography instrument (Agilent 7890 with an FID and a TCD detector). First, 50 mg of sample was loaded into the stainless-steel reactor. The reaction was carried out in a feed gas with a composition of 2% H$_2$, 2% n-C$_4$H$_{10}$ and He as carrier gas, and a gas hour space velocity (GHSV) of 18,000 mL·g$_{cat}$$^{-1}$·h$^{-1}$ on the basis of the whole feed gas (the total flow rate is 15 mL·min$^{-1}$).

The rate and conversion of n-butane and the selectivity of total C$_4$ olefin (n-butene and 1, 3-butadiene) were calculated by the following formula:

$$\text{n} - \text{butane Conversion} : \text{Conv.} = (\text{mol of the reacted})/(\text{mol of inlet n} - \text{butane}) \times 100\% \tag{1}$$

$$\text{Selectivity of C}_4 \text{ olefin} : \text{Selectivity}$$
$$= \{\text{mol of (butene formed} + 1, 3 - \text{butadiene formed})\}/(\text{mol of reacted}) \times 100\% \tag{2}$$

$$\text{n} - \text{butane rate} = (\text{flow rate of n} - \text{butane} \times \text{conversion of n} - \text{butane} \times 60)/$$
$$(\text{weight of Pt in the catalyst} \times 22.4) \tag{3}$$

The catalyst stability was described by a first-order deactivation model:

$$k_d = \{\ln[(1 - C_f)/C_f] - \ln[(1 - C_i)/C_i]\}/t \tag{4}$$

where $C_i$ is initial conversion after reaction 30 min; $C_f$ is final conversion value; t represents the reaction time (h); and $k_d$ is the deactivation rate constant (h$^{-1}$) that is used to evaluate the catalyst stability (the higher $k_d$ value is, the lower the stability).

**Computational details**. The Vienna ab initio simulation package (VASP) code[45,46] was used to perform spin-polarized DFT computations with the projector aug-mented wave (PAW) method[47,48]. The generalized gradient approximation in the form of the Perdew-Burke-Ernzerhof functional (PBE)[49] was chosen for electron exchange and correlation. An energy cutoff of 400 eV was employed for the plane wave expansion. The ground-state structure of bulk and surfaces were obtained by minimizing forces with the conjugate-gradient algorithm until the force on each ion is below 0.02 eV/Å, and the convergence criteria for electronic self-consistent interaction is 10$^{-5}$.

A model with Pt$_3$ cluster and Pt single atom embedded into a monovacancy at 5 × 5 supercell of graphene was adopted to simulate the active site of butane dehydrogenation (Pt$_3$-Gr) through comparative investigation between potential Pt cluster models and EXAFS data. The vacuum layer was set to 20 Å to avoid interaction from adjacent cells. The Monkhorst–Pack k-point set to 3 × 3 × 1 in the reciprocal lattice, and the electronic occupancies were determined according to the Gaussian smearing method with $\sigma = 0.1$ eV. Spin-polarized calculations were performed. For Pt (111) surface, a four-layer slab with a (3 × 3) supercell (Totally 36 atoms) was employed. The successive slabs were separated by a vacuum region as thick as 20 Å to eliminate periodic interactions. The Brillouin zone is sampled with a 3 × 3 × 1 k-points mesh by the Monkhorst–Pack algorithm. The electronic occupancies were determined according to the Methfessel-Paxton scheme with $\sigma = 0.2$ eV. The bottom two layers of the slab were kept fixed to their crystal lattice positions. Spin-polarization is not considered in Pt (111) calculation. We have calculated zero-point energies (ZPE) of reaction species and transition states.

The most stable configurations of the reactant and intermediates on Pt$_3$-Gr, Pt$_1$-Gr and Pt (111) surface were obtained by the standard minimization of density functional theory (DFT). These configurations were used as the initial states, from which the constrained optimization method as described by Plessow. P. N[50]. was used to search the transition states (TS). The TS optimization convergence was regarded to be achieved when the force on each atom was less than 0.05 eV/Å. All transition states have been verified to include only one imaginary harmonic frequency corresponding to the transition vector of the reaction. Furthermore, small distortions along the transition vector followed by optimization toward the minima verified the connectivity of the transition states. The entropy contributions of butane, 2-butene and hydrogen gas were included in the free energy calculations. The most important contributions arise from the translational entropy[51], which can be calculated using the following equation:

$$S = 1.5R\ln(2\pi MkT) - 3R\ln h + R\ln(kT/P) + 2.5R \tag{5}$$

where M, R, k, h, T, and P refer to the molecular weight, ideal gas constant, Boltzmann constant, Plank constant, temperature and pressure, respectively. In the energy diagrams shown in Fig. 3c, the free energies are reported at under conditions (723.15 K and 100 kPa), it was estimated that n-butane in the gas phase lost 1.34 eV of entropic energy (TS) in the adsorption; the desorption of 2-butene and H$_2$ gas gained 1.34 eV and 1.02 eV. It should be noted that the partial calculation work on the Pt$_3$-Gr model is based on the theoretical part of our previous work[35], and has been further improved in the Gibbs free energy calculations according to the main contribution of translational entropy.

## Data availability
The data supporting this article and other findings are available from the corresponding authors upon request. Source data are provided with this paper.

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

## Acknowledgements
This work was supported by the National Key R&D Program of China (2016YFA0204100, 2017YFB0602200), the National Natural Science Foundation of China (91845201, 21961160722, 22072162, 21703261, 21725301, 21932002, and 21821004), the Liaoning Revitalization Talents Program XLYC1907055, and the Sinopec China. N.W. hereby acknowledges the funding support from the Research Grants Council of Hong Kong (Project Nos. C6021-14E, N_HKUST624/19 and 16306818). The XAS experiments were conducted in Shanghai Synchrotron Radiation Facility (SSRF). D. M. thank OSSO State Key Lab for support.

## Author contributions
H.L. and D.M. conceived the research. X.Ch. conducted material synthesis and carried out the catalytic performance test. M.P., B.M., Y.D., and Zhe.J. conducted the X-ray

absorption fine structure spectroscopic measurements and analyzed the data. Y.C. and X.W. performed the DFT calculations. X.Ca. and N.W. contributed to the aberration-corrected high-angle annular dark-field scanning transmission electron microscopy. Zhi.J. performed some of the experiments. The manuscript was primarily written by X.Ch., D.X., H.L., and D.M. All authors contributed to discussions and manuscript review.

## Competing interests

The authors declare no competing interests.
