## [Peer Review File · Nature Communications]

REVIEWER COMMENTS

Reviewer #1 (Remarks to the Author):

The manuscript of Chen et al. studies the effect of using Sn promoters on the coordination number of platinum species supported on defect-rich graphene and the corresponding impact on the performance in the direct dehydrogenation of n-butane reporting the high performance of fully exposed Pt₃ species. Bimetallic Pt-Sn systems are widely studied for both the oxidative and direct dehydrogenation of small alkanes. Some of the authors have previously shown the possibility to obtain ‘fully’ dispersed Pt species with the assistance of tin on defect-rich graphene. The main novelty of this study relates to the deeper understanding attained, revealing a nonlinear relationship between the coordination number of Pt nanostructures and their activity, and identifying Pt₃ species as the optimal site. The experimental and theoretical analyses presented are conducted are of high quality, and the topic of resolving structure-performance relationships at the atomic level is timely and will be of broad interest. However, as outlined below, several important aspects need addressing before recommending this article for publication in Nature Communications.

Specific comments:

1. The introduction should identify the novel aspects of the study more clearly. Given the amount of work already done in the design of Pt catalysts for alkane dehydrogenation, including the author's previous work on Pt-Sn systems, a more critical discussion of the descriptors for the performance would be helpful to highlight the relevance of the current manuscript. Fundamentally, what is the reason for expecting a linear relation between the coordination number and the performance?
2. Experimental evidence for the formation of Pt₃ clusters and for their full exposure is not conclusive. All the techniques applied have limitations. The manuscript would be stronger if the authors recognized the challenges associated with characterizing these species, discussing the scope of both.
3. The difficulty in confirming the optimal structure experimentally could be compensated by a more systematic molecular-level description of the performance coming from DFT, but the theoretical insights are currently quite limited. Have the authors tackled how the Sn stabilizes the Pt species or its effect on the performance?
4. Similarly, although it is clearly difficult to synthesize metal species of different sizes in a controllable way, it would be valuable to provide theoretical insights for a series of samples (e.g. Pt atoms, dimers, trimers, tetramers) comparing the stability and reaction profiles.
5. As a minor comment, the change in nomenclature during the manuscript is confusing and unnecessary. Also, some of the data appear to be repeated (e.g. Fig. 3b and Fig. 4b)?

Reviewer #2 (Remarks to the Author):

The authors present a combined experimental and theoretical study of butane dehydrogenation catalyzed by supported Platinum catalyst. The DFT investigation focuses on Pt(111), single Pt atoms and Pt₃ clusters as active site motifs.

- 1) The conclusion from the experimental investigation (STEM) is that the cluster is only one monolayer thick. However, the computed Pt₃-cluster is standing upright and would be higher than a monolayer. Is this in agreement with the experimental observations?
- 2) The authors conclude that TS1 is rate-limiting, but energetically, dissociation of the products is actually more uphill in energy than TS1 for the Pt₃ cluster and Pt(111).
- 3) Instead of the energy (E) in Fig. 3c, the Gibbs free energy (G) profile needs to be analyzed to come to meaningful conclusions regarding rate-limiting step and kinetics.
- 4) For the investigation of catalysis, it is important that the most stable state (in terms of Gibbs free energy G) is located. The authors should describe (at least in SI) what they have investigated in this respect. For example, for Pt₃, many different isomers for “2H*” are likely to exist. Has the most stable one been found? Also, H₂ could desorb first, leading to C₄H₈*, which should also be investigated as a possible pathway, with a potentially more stable structure.
- 5) All computed total energies and Cartesian coordinates should be made available as SI.
- 6) The authors write that for Pt(111) they computed a 4-layer slab with 3x3-supercell and 48 atoms in total. That does not seem to make sense, as the total number of atoms should be 4x3x3=36. If the total number of atoms includes butane, then the sentence should be refined to make that more clear.

Reviewer #3 (Remarks to the Author):

The key sample in this paper, Pt_{1.7}Sn/ND@G, was published by this group in the literature last year (ACS Catal. 2019, 9, 7, 5998–6005). Many of the findings in this paper have already been reported including the entirety of Figure 1a-e, Figure 2c-d, Figure 3c and significant portions of Figure 4 (i.e. all of the characterization, catalytic activity, and DFT calculations on the Pt_{1.7}Sn/ND@G sample). As such, it is inappropriate to publish the manuscript as original work as it currently stands.

If the authors would like to remove the data on Pt_{1.7}Sn/ND@G and rewrite the paper accordingly, the manuscript could be considered for a catalysis-specific specialty journal.

Response to Reviewers

Reviewer #1:

The manuscript of Chen et al. studies the effect of using Sn promoters on the coordination number of platinum species supported on defect-rich graphene and the corresponding impact on the performance in the direct dehydrogenation of n-butane reporting the high performance of fully exposed Pt₃ species. Bimetallic Pt-Sn systems are widely studied for both the oxidative and direct dehydrogenation of small alkanes. Some of the authors have previously shown the possibility to obtain ‘fully’ dispersed Pt species with the assistance of tin on defect-rich graphene. The main novelty of this study relates to the deeper understanding attained, revealing a nonlinear relationship between the coordination number of Pt nanostructures and their activity, and identifying Pt₃ species as the optimal site. The experimental and theoretical analyses presented are conducted are of high quality, and the topic of resolving structure-performance relationships at the atomic level is timely and will be of broad interest. However, as outlined below, several important aspects need addressing before recommending this article for publication in Nature Communications.

Specific comments:

Comment 1: The introduction should identify the novel aspects of the study more clearly. Given the amount of work already done in the design of Pt catalysts for alkane dehydrogenation, including the author's previous work on Pt-Sn systems, a more critical discussion of the descriptors for the performance would be helpful to highlight the relevance of the current manuscript. Fundamentally, what is the reason for expecting a linear relation between the coordination number and the performance?

Response: We thank the reviewer for the nice suggestion. Following the suggestion, we have carefully revised our introduction part to describe the highlights of our current work and explain the reason for established relationship between the Pt structure (single atom, cluster and nanoparticle) and the n-butane DDH performance. The corresponding description was highlighted in the revised paper.

Comment 2: Experimental evidence for the formation of Pt₃ clusters and for their full exposure is not conclusive. All the techniques applied have limitations. The manuscript would be stronger if the authors recognized the challenges associated with characterizing these species, discussing the scope of both.

Response: We appreciate the reviewer for very nice comment. Surely, we need to say that the coordination number extracted from XAFS results are average coordination number. However, according to chemical adsorption experiments, the Pt_{1.7Sn}/ND@G catalyst with C.N._{Pt-Pt} around 2 has almost 100% dispersion which suggests that Pt species are fully exposed. But the reviewer is right. All the techniques have its limitations. The realistic Pt clusters on current catalyst could have distribution in both atomicity and configuration, which is also discussed in our recent prospective paper (doi:10.1021/acscentsci.0c01486). We have added a sentence in the manuscript to address the challenges associated with characterizing these species in the revised manuscript.

Comment 3: The difficulty in confirming the optimal structure experimentally could be compensated by a more systematic molecular-level description of the performance coming from DFT, but the theoretical insights are currently quite limited. Have the authors tackled how the Sn stabilizes the Pt species or its effect on the performance?

Response: We appreciate the reviewer for the suggestion. Sn is almost inert in alkane dehydrogenation. Interestingly, we found that Sn can be atomically dispersed on the ND@G support as shown in Figure R1. Meanwhile, no Pt–Sn bonding was observed in Pt_{1.7Sn}/ND@G, Pt_{3.4Sn}/ND@G and Pt_{6.8Sn}/ND@G catalyst before n-butane dehydrogenation as displayed in Figure 2 and Supplementary Figure S6. Actually, we have

investigated the structure and the local environment of Pt in spent $\text{Pt}_{1.7\text{Sn}}/\text{ND@G}$ catalyst (see Figure R2 and Table R1). In the Pt EXAFS, no Pt–Sn bonding was observed in spent $\text{Pt}_{1.7\text{Sn}}/\text{ND@G}$. These results suggest the structure of Pt_3 cluster was unchanged and no Pt–Sn alloy formed during the reaction. Thus, the atomically dispersed Sn species could only act as a geometric partition agent which can maintain highly dispersed Pt cluster structure on the ND@G support. The similar PtSn structure is also reported in Pt–Sn zeolite catalyst, in which Sn atoms only act as a geometric partition agent and do not directly interact with the Pt (*Nat. Mat.* **2019**, 18, 866–873; *Nat. Cat.* **2020**, 3, 628–638).

Figure R1 HAADF-STEM images of the Sn/ND@G catalyst with 0.5 wt% Sn.

Figure R2. Synchrotron XAFS measurements of spent $\text{Pt}_{1.7\text{Sn}}/\text{ND@G}$ (a) k^3 -weighted EXAFS spectra (b) the fittings of Pt L3-edge in k^3 space (c) Wavelet transform (WT) analysis

Table R1. Pt L3-edge EXAFS fitting results for spent $\text{Pt}_{1.7\text{Sn}}/\text{ND@G}$ catalyst

Sample	Shell	C.N.	$R(\text{Å})$	$\Delta E_0(\text{eV})$	S_0^2	R-factor
$\text{Pt}_{1.7\text{Sn}}/\text{ND@G}$	Pt–Pt	2.4	2.73	5.68	9.15	0.02
	Pt–O/C	2.4	1.95	5.68	8.13	
	Pt–Sn	-	-	-	-	-

Comment 4: Similarly, although it is clearly difficult to synthesize metal species of different sizes in a controllable way, it would be valuable to provide theoretical insights for a series of samples (e.g. Pt atoms, dimers, trimers, tetramers) comparing the stability and reaction profiles.

Response: We thank the reviewer for the good point. Theoretical calculation for comparison the series of samples (e.g., Pt atoms, dimers, trimers and tetramers) is timely and needful in DDH. From theoretical calculation, we are preparing another paper to systematically investigate the relationship between structure (e.g., single atoms, dimers, trimers and tetramers) and catalytic performance.

Comment 5: As a minor comment, the change in nomenclature during the manuscript is confusing and unnecessary. Also, some of the data appear to be repeated (e.g., Fig. 3b and Fig. 4b)?

Response: We thank the reviewer for pointing this out for us. Following the suggestion of the reviewer, we redefined the Pt_n/NDG , $Pt_3/ND@G$ and $Pt_1/ND@G$ as $Pt/ND@G$ (Pt NP), $Pt_{1.75n}/ND@G$ (Pt_3 cluster) and $Pt_1/ND@G$ (Pt SA) in Figure 4a and Figure 4b, respectively.

Reviewer #2:

The authors present a combined experimental and theoretical study of butane dehydrogenation catalyzed by supported Platinum catalyst. The DFT investigation focuses on Pt (111), single Pt atoms and Pt_3 clusters as active site motifs.

Comment 1: The conclusion from the experimental investigation (STEM) is that the cluster is only one monolayer thick. However, the computed Pt_3 -cluster is standing upright and would be higher than a monolayer. Is this in agreement with the experimental observations?

Response: Thanks for the good question. The claim of monolayered Pt clusters from the STEM investigation means that there are no Pt atoms covering each other and all Pt atoms in the clusters can get exposed to reactants during reaction, although we cannot obtain the exact height of each Pt atom from the 2D-projected STEM images. As illustrated in the perspective and top views of the DFT-calculated model in Figure R3 (a) and (b), respectively, the computed Pt_3 cluster has no Pt-Pt overlay in the thickness direction and thus contains single Pt layer. The simulated STEM image in Figure R3 (c) also shows no Pt-Pt overlaid contrast for the DFT model, in agreement with the experimental STEM observations in Figure R3 (d-f). We are sorry for our ambiguous writing brings much confusion to the reviewer. To clarify our “monolayered Pt clusters” conclusion more clearly, we corrected the writing and added more discussion in page 9 line 2.

Figure R3. Perspective (a) and top (b) views of the DFT-computed Pt_3 cluster model, respectively. Black balls for carbon while gray balls for platinum. (c) Simulated STEM image according to the model. The graphene support is invisible under our experimental conditions. HAADF-STEM images (d) of $Pt_{1.75n}/ND@G$, the clusters highlighted by yellow circles enlarged in (e) and (f).

Comment 2: The authors conclude that TS1 is rate-limiting, but energetically, dissociation of the products is actually more uphill in energy than TS1 for the Pt_3 cluster and Pt (111).

Response: According to our calculated experience, the Gibbs free energy of gas product at 450°C will fall significantly, which leads to the dissociation or desorption of gas product becoming much easier. Thus, we thought the TS1 was rate-limiting. However, as the reviewer mentioned below, the Gibbs free energy of reaction pathways can give more meaningful conclusions, which can take the effect of temperature into consideration in the reaction energy profile more clearly than we thought. Therefore, we have given the Gibbs free energy profile in the manuscript instead of the energy (E) profile in Figure 3c.

Comment 3: Instead of the energy (E) in Fig. 3c, the Gibbs free energy (G) profile needs to be analyzed to come to meaningful conclusions regarding rate-limiting step and kinetics.

Response: Instead of the energy (E) in Figure. 3c, the Gibbs free energy (G) profile has been given in the manuscripts, and one can see that the overall free energy barrier of butane dehydrogenation 1.19 eV for Pt_3 -Gr, 2.00 eV for Pt (111) and 2.52 eV for Pt_1 -Gr, respectively. The activity order based on the theoretical work is in agreements with our experimental results $(Pt_3\text{-Gr}) > Pt(111) > (Pt_1\text{-Gr})$.

Figure R4. The Gibbs free energy profile of butane dehydrogenation to 2-butene on the Pt₁-Gr, Pt₃-Gr and Pt (111). (purple: Pt₁-Gr, red: Pt₃-Gr, green: Pt (111)).

Comment 4: For the investigation of catalysis, it is important that the most stable state (in terms of Gibbs free energy G) is located. The authors should describe (at least in SI) what they have investigated in this respect. For example, for Pt₃, many different isomers for “2H*” are likely to exist. Has the most stable one been found? Also, H₂ could desorb first, leading to C₄H₈*, which should also be investigated as a possible pathway, with a potentially more stable structure.

Response: All the states of reaction intermediates was the most stable state in the manuscripts. Here, the other potential stable state of reaction intermediates has been listed in Supporting information (Supplementary Figure S15, S17 and S19). The reaction pathway of H₂ desorption firstly has been compared with the C₄H₈ gas desorption firstly as shown in Supporting information (Supplementary Figure S14, S16 and S18), which reveals the C₄H₈* will firstly desorb for those catalysts other than that the desorption of H₂ and 2-C₄H₈ on the Pt₁-Gr is comparative ($\Delta G = -0.96$ eV for H₂ desorption, $\Delta G = -1.11$ eV for 2-C₄H₈ desorption).

It should be noted that the most stable species in the energy profile for the three systems is the co-adsorbed C₄H₈*+2H* species, and its most stable isomer (or configuration) is the configuration noted by ‘a’ as shown in Supplementary Figure S17, which can be obtained by the dehydrogenation step from the ‘b’ configuration of C₄H₉*+H* species. Thus the ‘b’ configuration of C₄H₉*+H* species is used in the energy profile although it is not the most stable isomer of C₄H₉*+H* species (only higher 0.13 eV than the most one).

Comment 5: All computed total energies and Cartesian coordinates should be made available as SI.

Response: To check each intermediate more conveniently, all the computed reaction intermediates and the relative energies of different isomers has been given in SI.

Comment 6: The authors write that for Pt(111) they computed a 4-layer slab with 3x3-supercell and 48 atoms in total. That does not seem to make sense, as the total number of atoms should be 4x3x3=36. If the total number of atoms includes butane, then the sentence should be refined to make that more clear.

Response: We have checked again the theoretical method in the manuscripts, and found that the atom number of Pt (111) slab is 36 other than 48. Thanks very much for pointing out this mistake. We have fixed this mistake, and confirmed that the other part of theoretical methods is described correctly.

Reviewer #3:

The key sample in this paper, Pt_{1.7}Sn/ND@G, was published by this group in the literature last year (ACS Catal. 2019, 9, 7, 5998–6005). Many of the findings in this paper have already been reported including the entirety of Figure 1a-e, Figure 2c-d, Figure 3c and significant portions of Figure 4 (i.e. all of the characterization, catalytic activity, and DFT calculations on the Pt_{1.7}Sn /ND@G sample). As such, it is inappropriate to publish the manuscript as original work as it currently stands. If the authors would like to remove the data on Pt_{1.7}Sn/ND@G and rewrite the paper accordingly, the manuscript could be considered for a catalysis-specific specialty journal.

Response: We thank the reviewer for the nice comment. Our previous paper (ACS Catal. 2019, 9, 7, 5998–6005) provides a simple method for fabricating Pt₃ clusters. But we need to say that Pt loading was 1 wt% in our previous paper and it is 0.5 wt% in this current manuscript. Therefore, these two PtSn catalysts are totally different in Pt weight loading at least. Surely, we can obtain a similar Pt₃ structure in the Pt_{1.7}Sn/ND@G catalyst with 0.5 wt% by our previous reported method. Therefore, the data (including Figure 1a-e, Figure 2c-d, Figure 3c and significant portions of Figure 4) for the Pt_{1.7}Sn/ND@G catalyst with 0.5 wt% Pt loading is distinguished from the findings in our previous paper. Moreover, considering the industrial application, lower weight loading of noble metal is indispensable.

In our previous paper, we highlight the construction of unique structure Pt₃ and focus on comparing the performance differences between PtSn/ND@G catalyst and traditional Pt₃Sn/Al₂O₃ catalyst. However, in the current manuscript, we fabricated atomically dispersed Pt₃ clusters and precisely tune the coordination number (CN) of supported Pt clusters based our previous reported methods. The Pt₃ clusters with 0.5 wt% Pt loading showed higher conversion for n-butane DDH than Pt NPs and Pt SAs supported on ND@G. Moreover, we systemically established the structure–performance relationship by correlating the DDH activity with the average CN of Pt-Pt bond on ND@G supported Pt NP, Pt cluster, and Pt SA catalysts. The topic of resolving structure-performance relationships at the atomic level over supported Pt catalyst is timely and will be of broad interest in the DDH field. The revealed nonlinear relationship between CN of Pt nanostructures and DDH activity can provide a valuable insight for designing heterogeneous DDH catalysts in sub-nanoscale.

Special thanks to the reviewer for his/her insight comments, which really help us a lot in improving the manuscript.

REVIEWERS' COMMENTS

Reviewer #1 (Remarks to the Author):

The authors have taken appropriate action to the reviewers comments or provided appropriate justification for not doing so. Importantly, they have indicated that the synthetic procedure was previously developed, noted the limitations of EXAFS, and have extended the mechanistic discussion on the enhanced performance of the Pt₃ trimer. It would have greatly strengthened the article to provide further calculations there, but I realise that this is not a trivial task. Overall, the work is timely and I'm sure will be of interest to the alkane dehydrogenation community.

Reviewer #3 (Remarks to the Author):

Given that the only difference in the key catalytic sample between this manuscript and the authors previous publication is the total weight loading on the support (0.5 wt.% vs. 1 wt.%), and the structure of the active site appears to be otherwise identical based on their structural characterization, I stand by my previous comment that this manuscript is not suitable for publication in a journal such as Nature Communications. A follow-up study of this type would be more suitable for a catalysis-specific specialty journal.

Response to reviewers' comments

Reviewer #1

The authors have taken appropriate action to the reviewers comments or provided appropriate justification for not doing so. Importantly, they have indicated that the synthetic procedure was previously developed, noted the limitations of EXAFS, and have extended the mechanistic discussion on the enhanced performance of the Pt₃ trimer. It would have greatly strengthened the article to provide further calculations there, but I realise that this is not a trivial task. Overall, the work is timely and I'm sure will be of interest to the alkane dehydrogenation community.

Response: Special thanks to the reviewer's suggestions. It is a very good suggestion for strengthening this article. In the following work, we would further investigate the relationship between structure and catalytic performance with systematically calculations and in situ characterization techniques.

Reviewer #3

Given that the only difference in the key catalytic sample between this manuscript and the authors previous publication is the total weight loading on the support (0.5 wt.% vs. 1 wt.%), and the structure of the active site appears to be otherwise identical based on their structural characterization, I stand by my previous comment that this manuscript is not suitable for publication in a journal such as Nature Communications. A follow-up study of this type would be more suitable for a catalysis-specific specialty journal.

Response: We thank the reviewer for the nice comment. But we need to say that we fabricated atomically dispersed Pt₃ clusters and precisely tuned the coordination number of supported Pt clusters in the current manuscript. Moreover, we systemically established the structure–performance relationship by correlating the DDH activity with the average coordination number of Pt–Pt bond. The topic of resolving structure–performance relationships at the atomic level over supported Pt catalyst is timely and will be of broad interest in the DDH field.